# Environmental Factors Affecting the Diversity and Composition of Environmental Microorganisms in the Shaoxing Rice Wine Producing Area

**DOI:** 10.3390/foods12193564

**Published:** 2023-09-26

**Authors:** Qi Peng, Lili Zhang, Xiaoli Huang, Jianjiang Wu, Yujun Cheng, Guangfa Xie, Xinxin Feng, Xueping Chen

**Affiliations:** 1School of Life and Environmental Sciences, Shaoxing University, Shaoxing 312000, China; dr_pengqi@163.com (Q.P.); z18958809957@163.com (L.Z.); fxx_18066292357@163.com (X.F.); 15712657521@163.com (X.C.); 2Shaoxing Testing Institute of Quality and Technical Supervision, Shaoxing 312000, China; tqntqn56@126.com (X.H.); cs8033197@126.com (J.W.); yujun880106@sina.com (Y.C.); 3Laboratory of Pollution Exposure and Health Intervention of Zhejiang Province, College of Biology and Environmental Engineering, Zhejiang Shuren University, Hangzhou 310015, China

**Keywords:** Illumina MiSeq sequencing, fermented beverage, bacterial communities, food environment, fungal community, environment

## Abstract

Shaoxing rice wine is a notable exemplar of Chinese rice wine. Its superior quality is strongly correlated with the indigenous natural environment. The results indicated that Firmicutes (75%), Actinobacteria (15%), Proteobacteria (5%), and Bacteroidetes (3%) comprised the prevailing bacterial groups. Among the main bacterial genera, *Lactobacillus* was the most abundant, accounting for 49.4%, followed by *Lactococcus* (11.9%), *Saccharopolyspora* (13.1%), *Leuconostoc* (4.1%), and *Thermoactinomyces* (1.1%). The dominant fungal phyla were Ascomycota and Zygomycota. Among the dominant genera, *Saccharomyces* (59.3%) prevailed as the most abundant, followed by *Saccharomycopsis* (10.7%), *Aspergillus* (7.1%), *Thermomyces* (6.2%), *Rhizopus* (4.9%), *Rhizomucor* (2.2%), and *Mucor* (1.3%). The findings demonstrate that the structure of the bacterial and fungal communities remains stable in the environment, with their diversity strongly influenced by climatic conditions. The continuous fluctuations in environmental factors, such as temperature, air pressure, humidity, rainfall, and light, significantly impact the composition and diversity of microbial populations, particularly the dominant bacterial community.

## 1. Introduction

Huangjiu, a Chinese national specialty commonly referred to as rice wine, is a prominent member of the world’s three major brewing wines, along with wine and beer, holding a significant position. The intricate brewing process involving rice soaking, steaming, fermentation, pressing, sterilization, storage, and aging, combined with the unique climatic conditions (including temperature, humidity, and precipitation), which results in a distinct and intricate microbial community structure within Shaoxing rice wine. This complex ecosystem contributes to the unique flavor profile of Shaoxing rice wine [1,2].

Shaoxing is situated on the southeastern coast of China (29°13′35′′~30°17′30′′ N, 119°53′03′′~121°13′38′′ E), with Kuaiji Mountain to the south and Yushao Plain to the north. Due to the influence of external ocean currents and the barrier of Siming Mountain and Kuaiji Mountain, the producing area has formed a relatively stable and humid environment in the outer sea. The average annual temperature in the producing area ranges from 16.2 °C to 16.5 °C (the seasonal temperature ranges: from 15.7 °C to 15.9 °C in spring; 28.2~28.7 °C in summer; 17.7~18.3 °C in autumn; and 3.9~4.4 °C in winter) The annual humidity (the degree of saturation of the distance between water vapor in the air) is 77% to 82%, and the annual precipitation is 1301 to 1465 mm. This unique geographical environment is suitable for the growth of brewing microorganisms.

The microbial community is unevenly distributed in the environment and exhibit biogeographical patterns. In large organisms, these biogeographic patterns are determined via biological factors such as competition between species, predation, and parasitism, as well as abiotic factors such as temperature range, nutrients, and biodistribution [3].

Shaoxing rice wine fermentation involves a complex microbial community that exhibits symbiotic interactions [4]. This community strongly relies on its environment and contributes to the flavor and taste of the wine to some extent. The fermentation process follows a co-fermentation mode of mixed strains in an open environment [5]. Numerous studies have highlighted that the dominant fungi in rice wine starter cultures (such as wheat Qu and Jiuyao) are mainly mold, followed by yeast [6]. In terms of bacterial microorganisms, *Bacillus*, *Staphylococcus*, *Leuconostoc*, *Pediococcus*, *Lactobacillus*, and *Lactococcus* are predominant [7]. The microbiota involved in the Shaoxing rice wine brewing process primarily encompasses fungi (such as *Saccharomyces*, *Aspergillus*, *Rhizopus*, and *Monascus*) and bacteria (including *Bacillus* and *Lactobacillus*, such as *Lactobacillus plantarum*, *Lactobacillus brevis*, *Micrococcus pentosus*, and *Micrococcus yeast*) [4,8]. Microbial community diversity is influenced by factors such as the microbial inoculum, processing technology, production environment, and climatic conditions [9]. Currently, micro-fermentation morphology attracts significant attention from rice wine researchers, particularly the interplay between environmental microecology and the microorganisms involved in rice wine fermentation.

This study utilizes high-throughput sequencing technology to investigate the microbial community structure and the diversity within the core area of the Shaoxing rice wine production area throughout the four seasons of a year. Additionally, correlation analysis is employed to analyze the impact of environmental factors on these microbial communities. The findings of this study offer a theoretical foundation for further understanding the connection between fermented ecological resources and the quality of fermented foods.

## 2. Materials and Methods

### 2.1. Sample Collection

All environmental samples were obtained from 10 wineries producing Shaoxing rice wine, a product protected via geographical indications. Samples were collected in the four seasons from the brewing tools, indoor windowsills, door frames, floors, and areas around the wine vats [9]. Representative samples were taken five times in each season, and 10 groups of the samples were randomly mixed in equal quantities each time (a total of 200 groups of the samples). The fresh samples were sealed and put into the refrigerator at −80 °C for low-temperature storage until use.

### 2.2. High-Throughput Sequencing Analysis

#### 2.2.1. Microbial Enrichment in Pretreatment Samples

Sample pretreatment was carried out on the basis of the previous studies by Zhu Y et al. [10]. The mixed sample of 200 mg, 1 mL of ethanol (70%), and 4 to 5 glass beads were added into the 200 mL centrifuge tube, shook for 5 min, centrifuged at 10,000 rmp for 3 min, and then the upper liquid was discarded. A phosphate (PBS) buffer at 10,000 rmp was added to the lower liquid and centrifuged for 3 min, then the supernatant was discarded. The 200 mL centrifuge tubes were placed on absorbent paper until no liquid remained.

#### 2.2.2. DNA Extraction and PCR Amplification

Genomics DNA from the samples were extracted via MP Fast DNA SPIN Kit for Soil Kit (MP Biomedical). Qubit 3.0 fluorometer was used to determine the concentration of DNA, and a 0.8% agar gel electrophoresis buffer was used to detect DNA integrity [10].

The V3 and V4 regions of bacterial 16S rRNA and fungal ITS rRNA were amplified based on previous research in our laboratory [11].

The PCR amplification system and conditions have been modified according to previous methods [12]. PCR reactions were performed in triplicate using 0.4 μL of FastPfu polymerase, 4 μL of 5 × FastPfu buffer, 0.8 μL (5 μM) of each primer, 2 μL of 2.5 mM dNTPs, and 10 ng of template DNA under the following conditions: 95 °C for 3 min, 27 cycles at 95 °C for 30 s, 55 °C for 30 s, and 72 °C for 45 s, and a final extension at 72 °C for 5 min.

#### 2.2.3. Data Processing and Analysis

Sequencing was performed on the Illumina MiSeq sequencing platform (Illumina in San Diego, CA, USA and Majorbio Technology Co., Ltd. in Shanghai, China). The data were analyzed on the QIIME platform and Operational Taxonomic Units (OTU) were identified (97% similarity) [13]. OTU is classified via the Ribosome Database Program (RDP) classifier [14]. Alpha diversity (the Number of OUT, Shannon index, Coverage, Simpson index, and Chao1) indicates the richness and diversity of the microbial communities.

The Shannon index and the Simpson index indicate the degree of diversity of the community.

### 2.3. Environmental Factors

The data of temperature, humidity, precipitation, light, pressure, air quality, and other environmental factors were provided by the Environmental Monitoring Center of Shaoxing City.

### 2.4. Statistical Analysis

To explore and visualize the relationships between Alpha diversity, the dominant environmental microbial (fungi, bacteria), and the environmental factors, a correlation matrix was constructed by calculating all possible pairwise Pearson’s rank correlations at the phylum and genus levels. A Pearson’s test was performed using the SPSS software version 18.0 (SPSS Inc., Chicago, IL, USA). A Pearson’s correlation coefficient (ρ) > 0.5 (or less than −0.5) was performed and a statistical significance of *p* < 0.01 was set to indicate a co-occurrence (or negative) event for a particular correlation.

## 3. Results and Discussions

### 3.1. Correlation between Environmental Factors and Alpha Diversity of Bacterial and Fungal Communities

The study utilized the Illumina MiSeq platform to analyze the microorganisms in the environmental samples. The OTUs, with a 97% similarity, were clustered using the USEARCH algorithm. Several metrics, including the number of OTUs, Simpson index, Coverage, Chao1, and Shannon index, were employed to assess the alpha diversity of bacteria and fungi in the environment (Figure 1) [15]. Across all samples, a total of 5780 OTUs were identified for bacteria, with an average of 289 OTUs per sample. Similarly, a total of 1379 OTUs were found for fungi, with an average of 69 OTUs per sample. In terms of seasonality, the number of bacterial OTUs ranged from 176 to 391, while fungal OTUs ranged from 63 to 75 (Figure 1A).

The results indicated significant differences in the diversity of environmental microorganisms (fungi and bacteria) between autumn (September to November) and winter (December to February). The diversity of the bacterial community is much greater than that of the fungal community during Huangjiu brewing [2,7]. Figure 1B–D illustrates that during autumn, the Chao1, Shannon index, and Simpson index were all at their lowest, indicating a reduced diversity and richness of environmental microorganisms. Conversely, the situation reversed during winter. Additionally, the most significant changes in the environmental microbial diversity occurred during summer, with notable variations among the different sample groups. The coverage of all samples exceeded 99.9% (Figure 1E), suggesting that the sequencing results accurately reflected the actual state of the samples [15]. These observations indicate that the growth and enrichment of the microbial communities were unfavorable in autumn but favorable in winter. Furthermore, the fluctuating microbial community diversity in summer aligns with the susceptibility of rice wine to spoilage during that season.

A correlation analysis was conducted to examine the relationship between the environmental factors and the microbial diversity, including fungi and bacteria. The results of this analysis can be seen in Figure 2 and Figure 3. The Pearson correlation coefficient was used to divide the data into three groups, representing Alpha diversity and environmental factors. Bacterial diversity in both summer and winter showed significant correlations with environmental factors. During the summer, the number of operational taxonomic units (OTUs) was positively correlated with humidity and the eight-hour average concentration of ozone (O3_8h), but negatively correlated with temperature, precipitation, and light. In the winter, the number of OTUs was positively correlated with temperature, air pressure, humidity, and precipitation, but negatively correlated with sunlight. Bacterial communities showed a significant correlation with temperature in all four seasons. Specifically, the correlation was positive in spring, autumn, and winter, but negative in summer.

Unlike bacteria, fungal community diversity was only related to certain environmental factors. In spring, the Simpson index showed a negative correlation with temperature, but a positive correlation with precipitation. In summer, Alpha diversity was negatively correlated with sulfur dioxide (SO2), which is commonly used as a chemical preservative in the wine industry to prevent the growth of spoilage microorganisms [16], whereas summer is a high season for spoilage microorganisms [17]. In winter, the Shannon index showed a positive correlation with temperature, humidity, and nitrogen dioxide (NO2).

The results indicated that multiple environmental factors in the four seasons influenced the diversity of bacteria and fungi in the environment of the yellow rice wine producing areas and played a regulatory role. Temperature, in particular, had a significant effect on the regulation of the microbial community. The correlation between environmental factors and the Alpha diversity of the bacterial and fungal communities varied across different seasons. The microbial communities are regulated via environmental factors in different seasons, leading to a substantial difference in the microbial diversity between the four seasons. The level of dependence of microorganisms on their own growth resources and environment differs, and the symbiotic relationship between them further explains the variations in environmental microbial community diversity across different periods.

### 3.2. Similarity Analysis of Bacterial and Fungal Community Structure

Based on Illumina MiSeq sequencing data, a principal component analysis (PCA) was conducted to assess the similarity and difference in the bacterial community structure across different seasons (Figure 4A). The PCA results revealed that PCA1 and PCA2 collectively accounted for 72.34% of the variation in the environmental bacterial community. The seasons could be classified into four distinct categories, with the bacterial communities in spring and winter exhibiting a close resemblance, suggesting minimal disparity between these two seasons. Conversely, the bacterial community structure in summer differs significantly from the other three seasons, as exemplified by the greater distance separating it from the rest. A possible explanation for this dissimilarity may be the higher summer temperatures in the Shaoxing rice wine region (ranging from 28.2 to 28.7 °C), which likely exert a substantial influence on the bacterial community. Moving on to the fungal community, the principal component analysis (Figure 4B) elucidated that PCA1 and PCA2 collectively explained 74.56% of the variation in the environmental fungal community. Interestingly, the fungal communities in spring and autumn were adjacent and exhibited some overlap, indicative of the minimal differences in their community structures. In contrast, there was a substantial dissimilarity in the fungal community structure between summer and winter, as evidenced via the extensive distance separating the two seasons.

### 3.3. Structural Changes of Microbial Colonies in Environmental Samples

#### 3.3.1. Bacterial Community Structure

Based on the sequencing data, the regional sequences of bacterial 16S rRNA gene V3-V4 and fungal ITS rRNA gene ITS1-ITS2 in the four seasons were analyzed, which accurately reflected the complete species and structure of bacteria and fungi in the environment. At the phylum level, the composition of the bacterial community structure is shown in Figure 5A. In the four seasons, a total of 15 phyla were detected. At the phyla level, the main bacteria were Firmicutes, Cyanobacteria, Actinobacteria, Proteobacteria, and Bacteroidetes. Among them, the abundance of Firmicutes was the highest, especially in summer, with the highest relative abundance of 85.5% and the average of 81.6%. At the phylum level, the second dominant bacteria were Actinobacteria with an average relative abundance of 10.1% in summer. In the other seasons, its relative abundance remained between 15.2% and 17.5%. Other dominant bacteria were Cyanobacteria (1.6%), Proteobacteria (5.4%), and Bacteroidetes (7.8%). Except for dominance, the remaining 10 phyla were all less than 0.1% in relative abundance in any sample and were classified as other. Numerous past studies have shown that microorganisms play a key role in the production of fermented food [12,18,19].

The findings indicated that the prevalent bacterial phyla in the environment throughout all four seasons were Firmicutes, Actinobacteria, Proteobacteria, and Bacteroidetes. These bacterial phyla exhibited relative stability in their dominance within the environment. Firmicutes, the largest bacterial phylum, encompasses several significant fermentation bacteria found in rice wine, such as *Lactobacillus* and *Leuconostoc*. These bacterial communities exert inhibitory effects on other microorganisms, enabling the preservation of normal fermentation in fermented foods and providing protection against pathogen interference during the fermentation process [20].

The diversity of environmental microorganisms was assessed based on the genus structure [21]. A total of 505 bacterial genera were identified from the environmental samples (Figure 5B). In the four seasons, 452, 486, 393, and 442 bacterial genera were detected, respectively. The top 16 genera were consistently detected in all seasons, comprising 93.6–95.6% of the total relative abundance. *Lactobacillus* exhibited the highest relative abundance in all seasons, accounting for 38.5% in spring, 71.4% in summer, 44.3% in autumn, and 43.3% in winter. The second most dominant genus was *Saccharopolyspora*, with a relative abundance of 16.0% in winter and 9.5% in summer. Apart from *Lactobacillus* and *Saccharopolyspora*, the relative abundance of *Lactococcus* and *Leuconostoc* in the environmental bacterial communities was also noteworthy, ranging from 0.06% to 24.0% and from 0.02% to 8.4%, respectively, at the genus level. The relative abundance of *Thermoactinomyces* peaked at 1.8% in summer and reached its lowest point of 0.5% in winter. Other genera accounted for a mean relative abundance of 6.4% in summer and 4.4% in winter. Previous research demonstrated that *Lactobacillus*, *Saccharopolyspora*, *Leuconostoc*, *Lactococcus*, *Staphylococcus*, *Bacillus*, *Weissella*, *Pseudomonas*, *Thermoactinomyces*, and *Enterobacteria* were the most predominant bacteria in Shaoxing rice wine and Shanghai rice wine at the genus level, aligning with the findings of this study [22].

These results showed that the relative abundance of the bacterial communities varies with the seasonal environmental factors. The most miscellaneous bacteria were found in summer and the least in winter, which may explain why rice wine brewed in summer is more prone to rancidity and deterioration while that brewed in winter is of better quality. In conclusion, the growth and enrichment of bacteria were significantly affected by climatic conditions. *Lactobacillus*, *Lactococcus*, and *Pediococcus* belong to Lactic acid bacteria (LAB), and the relative abundance of LAB in all seasons is 62.5%, 76.3%, 59.9%, and 55.6%, respectively. LAB has an inhibitory effect on other microorganisms [15], and it can convert sugar into lactic acid. Lactic acid can react with ethanol to produce ester substances, which is an important flavor substance in the fermentation process of rice wine [12]. Enterobacteriaceae can degrade sugars via glycolysis and pentose phosphate to produce organic acids.

It is worth noting that the number of acid-producing bacteria is higher in summer than in the other seasons, especially with a particularly high percentage of *Lactobacillus*, which includes *Lactobacillus brevis*. Studies have shown that *Lactobacillus brevis* can lead to rancor in wine and beer and produce a pungent odor [17]. *Lactobacillus brevis* has strong acid-producing capacity, and an excessive content easy leads to the rancidity of rice wine, which affects the final quality of rice wine. Low-temperature conditions in winter inhibit the growth and reproduction of *Lactobacillus brevis*, and yellow rice wine is not easy to induce rancidity. At the same time, *Saccharopolyspora* and *Leuconostoc* were higher in winter than in the other seasons, which may be due to the low temperature and dry environmental conditions, which reduces the enzyme activity and slows down the metabolic rate of microorganisms, which contributes to the slow growth of microorganisms in rice wine brewing. It promotes the stability of the ecosystem, the subsequent aging of rice wine, and the synthesis of flavor substances. The content of miscellaneous bacteria in the winter environment is less than that in the other seasons, and the comprehensive environment in winter is more conducive to the brewing of high-quality rice wine [23,24]. 

#### 3.3.2. Comparison of Fungal Community Structure

In the earliest fermentation process, owing to the sufficiency of nutrients and a favorable environment, fungal microorganisms grow rapidly and multiply to form a complex fungal community. At the phylum level, a total of eight phyla were detected in the environmental samples, mainly Ascomycota and Zygomycota (the average content was more than 1%) (Figure 5C). The relative abundance of Ascomycota remained in the range of 81–91.7% in the four seasons, accounting for the vast majority of the fungal community in the whole environmental sample, and it was the first dominant fungal phylum in the whole process of rice wine brewing. A large number of studies have shown that Ascomycota is the dominant fungus not only in the brewing process of rice wine, but also in fermented foods such as white wine and edible vinegar [25]. Zygomycota is the second dominant fungus phylum, which includes the common fungus genera in rice wine wheat Qu and Jiuyao, such as *Rhizopus*, *Rhizomucor*, and *Mucor* [26].

A total of 93 fungi were detected at the genus level in the environmental samples. The seven main fungi genera are *Saccharomyces*, *Rhizopus*, *Thermomyces*, *Aspergillus*, *Saccharomycopsis*, *Rhizomucor*, and *Mucor* (Figure 5D). In the environmental samples, although the species of fungi were similar in the four seasons, the relative abundance of some fungi varied greatly at the genus level. For example, the relative abundance of *Rhizopus* is 11.3% in spring and 0.4% in summer. The *Thermomyces* relative abundance is 14.4% in summer and only 0.2% in winter. The relative abundance of *Saccharomyces* was the highest in the four seasons. *Saccharomyces* was the main dominant fungus genus in the environment of the rice wine producing areas. The content was 55.2% in spring, 63.2% in summer, 59.8% in autumn, and 59.2% in winter. Yeast is an important fungus in fermented wine and plays a leading role in ethanol production [27]. In the brewing process of Shaoxing rice wine, yeast can activate the enzyme system in the body of microorganisms to promote the conversion of carbohydrates to ethanol [20]. *Rhizopus* is the second most dominant genus of fungi. The relative abundance was 11.3% in spring, 0.4% in summer, 5.7% in autumn, and 2.0% in winter. *Rhizopus* has been widely used in the fermentation industry, which can inhibit the growth and reproduction of pathogenic bacteria and produce acid, thus aiding the fermentation of food [21].

### 3.4. Relationship between Environmental Factors and Dominant Microbial Communities

In order to analyze the dynamic relationship between the environmental factors and the environmentally dominant bacteria community, 27 dominant fungi and bacteria with an abundance ≥ 1.00% were selected as the research subjects. As shown in Figure 6, 23 dominant fungi and bacteria were significantly associated with one or more environmental factors. Due to the unknown that the different environmental factors act on the microbial communities at the same time, we also analyzed the correlation between the different environmental factors. The results show that temperature was negatively correlated with air pressure, positively correlated with precipitation, and negatively correlated with sunlight and humidity.

A total of 12 kinds of superiority bacterium (*Trichosporon*, *Weissella*, *Thermoactinomyces*, etc.) are associated with a significant temperature. Among them, the *Lactobacillus*, *Thermomyces*, *Weissella*, *Thermoactinomyces*, and *Trichosporon* were significantly positively related to temperature, and *Pichia*, *Saccharomycopsis*, and *Leuconostoc* had a significantly negative correlation with temperature. Since precipitation and temperature are positively correlated, the correlation between precipitation and the main microbial in the environmental samples is consistent with temperature. Eight main microbials (*Pichia*, *Thermoactinomyces, Thermomyces*, etc.) are significantly related to air pressure. Six main microbials (*Acetobacter*, *Acinetobacter*, *Mucor*, etc.) were significantly correlated with humidity, among which *Terribacillus*, *Acetobacter*, and *Acinetobacter* were significantly negatively correlated with humidity. Nine main microbials (*Trichosporon*, *Thermoactinomyces*, *Weissella*, etc.) were significantly related to precipitation. Ten main microbials (*Pichia*, *Acetobacter*, *Terribacillus*, etc.) were significantly correlated with sunlight, among which *Terribacillus*, *Acetobacter*, and *Acinetobacter* were significantly positively correlated with sunlight. 

These data indicated that temperature, precipitation, and sunlight were conditional factors for the micro-ecology and diversity of the environment. Previous studies have shown that temperature is an important factor in determining the quality of rice wine during fermentation [22,28,29]. High-quality rice wine is more suitable for room temperature or cold drinks because yeast is the main source of the flavor of rice wine, so it can be produced after long-term fermentation at low temperatures. *Thermoactinomyces* and *Thermomyces* are high-temperature-resistant microorganisms, which produce fatty acids, phosphatase, and thermostable enzymes respectively, and then degrade the carbohydrates to provide energy sources for the dominant bacteria in the brewing process [30]. At the same time, a higher storage temperature will easily lead to the deterioration of rice wine, and drinking rancid rice wine will lead to the increase in prevalence, so it is necessary to control the temperature during the brewing and storage of rice wine [31]. At the same time, there was a significant negative correlation between nitrogen dioxide and sulfur dioxide and yeast. Yeast is the main dominant fungus in the process of rice wine brewing. The air quality of Shaoxing city affects the brewing of rice wine to some extent. 

In conclusion, environmental factors play a decisive role in the regulation of the microbial community’s structure and diversity, and different environmental factors have a synergistic effect. Ambient temperature, air pressure, precipitation, sunlight, and humidity affect the growth and enrichment of microorganisms, indirectly or directly. The unique climatic conditions in Shaoxing provided suitable growth and reproduction conditions for rice wine fermentation microorganisms. These conditions allow the fermentation microorganisms to stably reproduce in the long-term and provide enrichment. At the same time, it once again proved that Shaoxing is a famous rice wine producing area.

## 4. Conclusions

This article uses high-throughput sequencing technology to study the structure and diversity of the environmental microbial communities in the Shaoxing rice wine production area. The dominant microbial (genus) composition was common and relatively stable across seasons. There were five dominant bacterial genera in the environmental samples: *Lactobacillus* (49.4%), *Lactococcus* (11.9%), *Saccharopolyspora* (13.1%), *Leuconostoc* (4.1%), and *Thermoactinomyces* (1.1%), among which *Lactobacillus* was the absolute dominant bacteria. There were seven major fungal genera: *Saccharomyces* (59.3%), *Rhizopus* (4.9%), *Thermomyces* (6.2%), *Aspergillus* (7.1%), *Saccharomycosis* (10.7%), *Rhizomucor* (4.9%), and *Mucor* (1.3%), among which *Saccharomyces* was the absolute dominant fungus.

Environmental factors, especially temperature, air pressure, precipitation, humidity, and sunlight, significantly affected the dynamic changes of the community structure and diversity of the environmental microorganisms, which was one of the key factors affecting the quality of Shaoxing rice wine.

The results of this study provide a reference for researchers to understand the contribution mechanism of fermentation microecology to fermentation quality and provide theoretical support for the development and support for the development and optimization of the fermentation process in Shaoxing rice wine.

## Figures and Tables

**Figure 1 foods-12-03564-f001:**
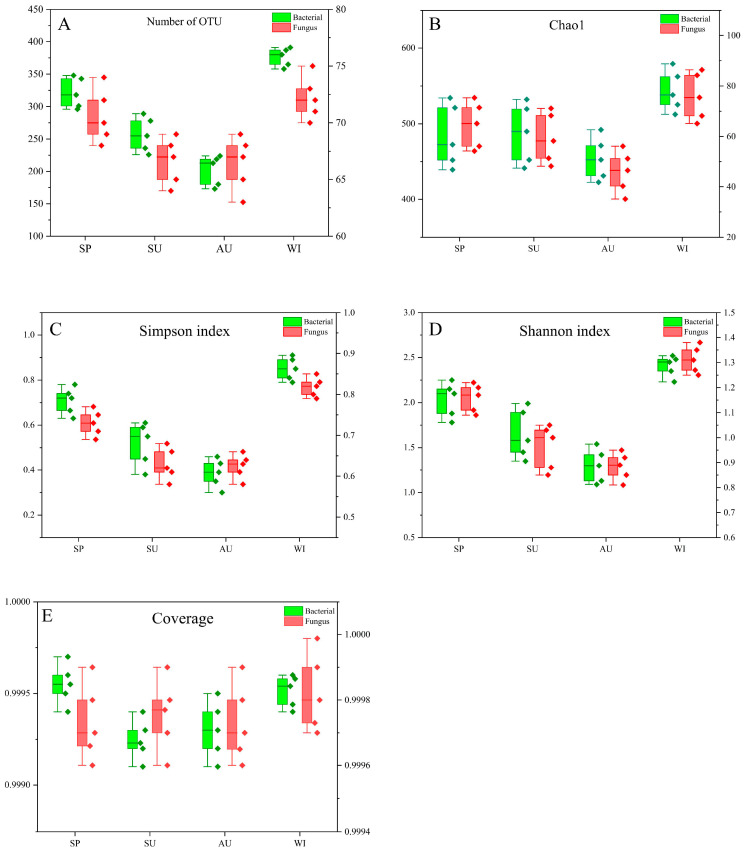
Alpha diversity of bacteria and fungi in different seasonal 508 environmental samples: (**A**) number of OTU; (**B**) Chao1; (**C**) Shannon index; (**D**) Simpson index; (**E**) coverage. Note. SP: Spring; SU: Summer; AU: Autumn; WI: Winter.

**Figure 2 foods-12-03564-f002:**
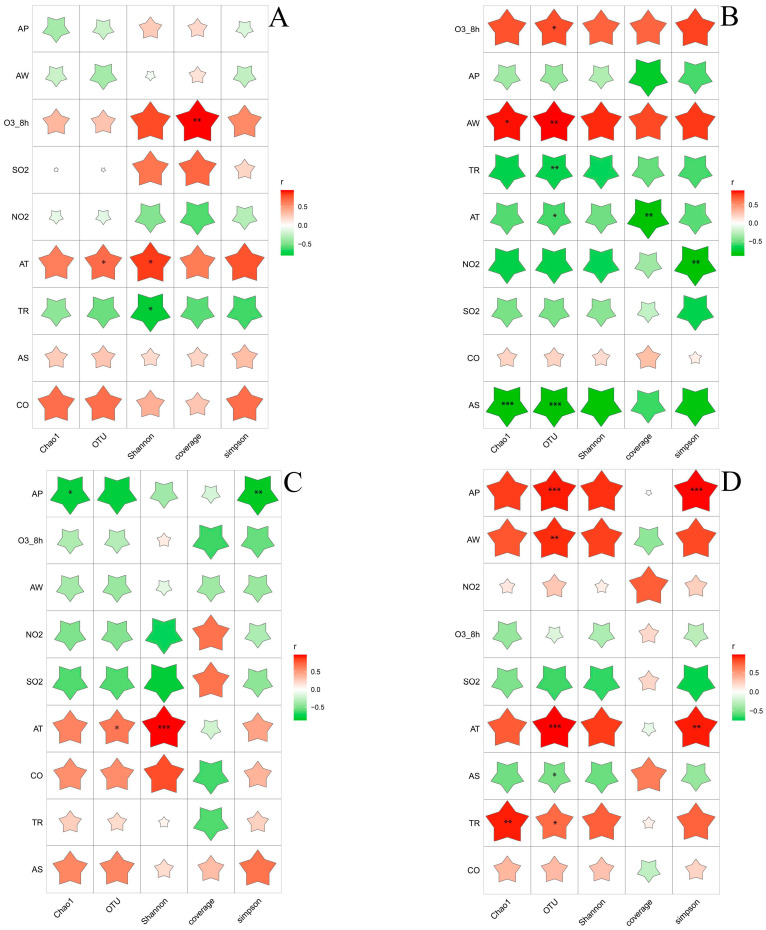
Heat map of Pearson correlation coefficient of bacterial 513 alpha diversity index and environmental factors. (**A**) Spring, (**B**) summer, (**C**) autumn, and (**D**) winter. The color of the star represents the direction of the correlation coefficient. *: 0.01 < *p* < 0.05, **: 0.001 < *p* ≤ 0.01, ***: *p* ≤ 0.001. Note. AT: Temperature; AP: Air pressure; AW: Humidity; TR: Precipitation; AS: Sunlight.

**Figure 3 foods-12-03564-f003:**
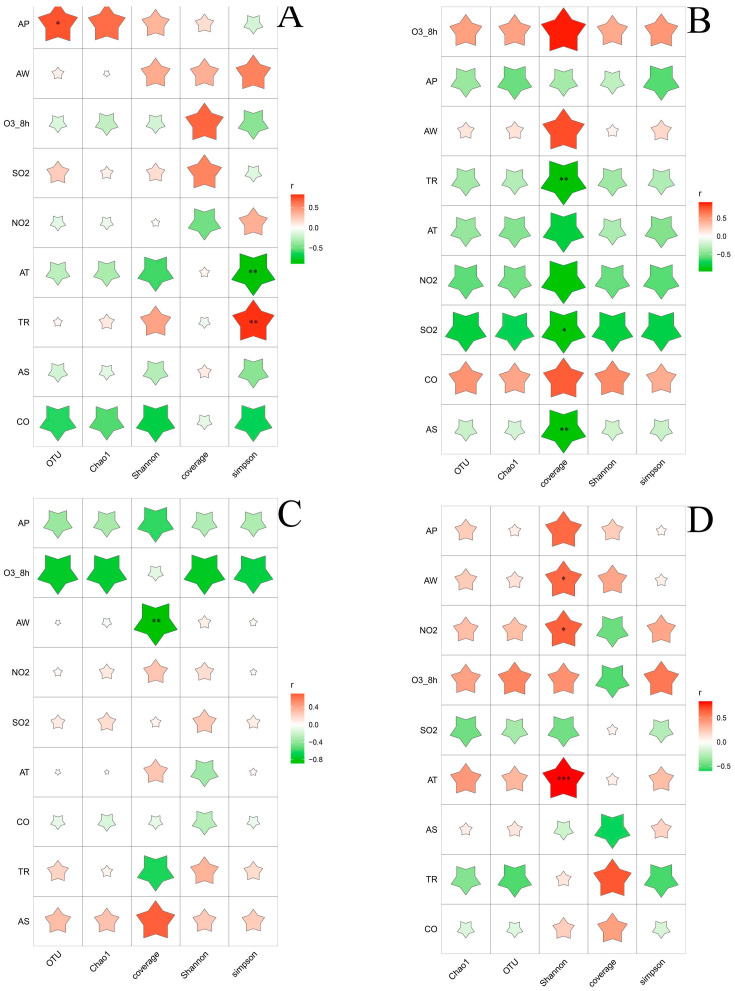
Heat map of Pearson correlation coefficient of fungal 520 alpha-diversity index and environmental factors. (**A**) Spring, (**B**) summer, (**C**) autumn, and (**D**) winter. The color of the star represents the direction of the correlation coefficient. *: 0.01 < *p* < 0.05, **: 0.001 < *p* ≤ 0.01, ***: *p* ≤ 0.001.

**Figure 4 foods-12-03564-f004:**
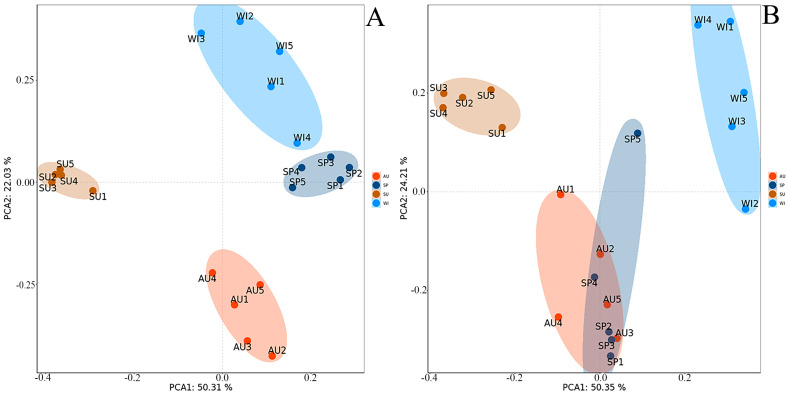
(**A**) Bacterial community PCA; (**B**) Fungal community PCA.

**Figure 5 foods-12-03564-f005:**
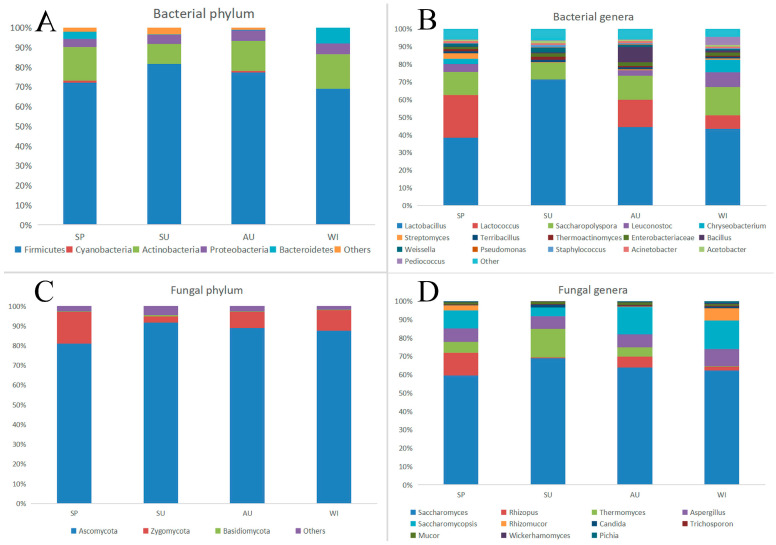
(**A**) Relative abundance of major bacterial communities at the phyla level (>0.1%). (**B**) Relative abundance of major bacterial communities at the genus level (>0.1%). (**C**) Relative abundance of main fungal communities at the phyla level (>0.1%). (**D**) Relative abundance of main fungal communities at the genus level (>0.1%). The horizontal coordinate represents the sample name, and the vertical coordinate represents the relative abundance.

**Figure 6 foods-12-03564-f006:**
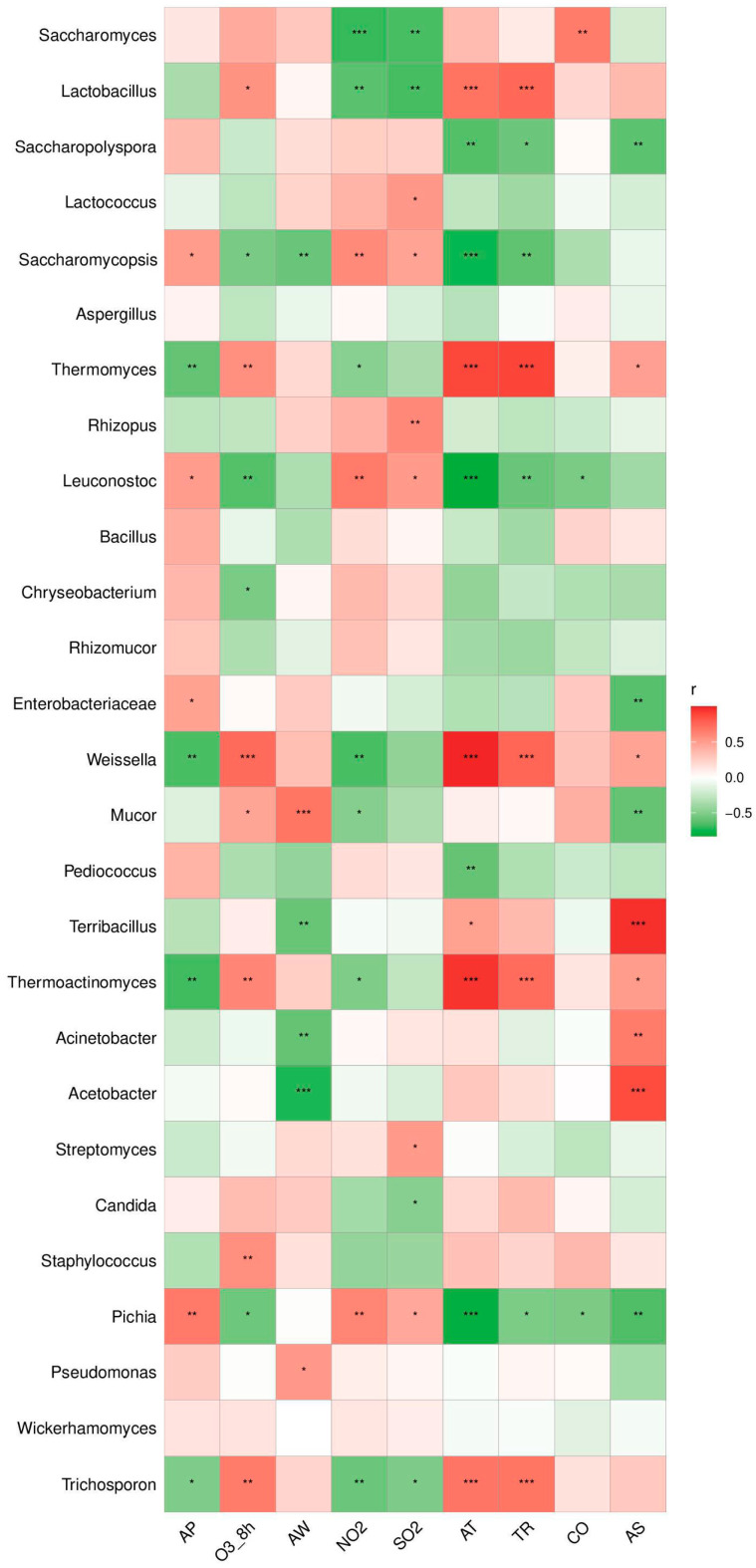
Heat map of Pearson’s rank correlation coefficient between relative abundance (average > 1%) of main microbial (bacteria and fungi) groups at genus level and environmental factors. The square color indicates the size and direction of the correlation coefficient. *: 0.01 < *p* < 0.05, **: 0.001 < *p* ≤ 0.01, ***: *p* ≤ 0.001.

## Data Availability

The data used to support the findings of this study can be made available by the corresponding author upon request.

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
