# Peer review of "Environmental Factors Affecting the Diversity and Composition of Environmental Microorganisms in the Shaoxing Rice Wine Producing Area"

_foods, 2023, doi:10.3390/foods12193564_

Round 1
Reviewer 1 Report
Overall, the manuscript shows insights into traditional rice wine. However, the manuscript must be improved. The authors need to revise the introduction to show the statements which support this study. Also, in the material and methods, the authors need to improve the description of the experiment conditions. In the results, there are many issues to review. The authors need to describe the main finding and emphasize the difference. (ie. How could the temperature interfere with the prevalence of different? Which genera was most prevalent in winter, and why can this phenomenon occur? Please, synthesize the principal results and discuss the influence o the parameters on index diversity.
Specific commentaries
Lines 30 -34: The genera are italic. Please, correct.
Keywords: Fermented beverage instead of Shaoxing rice wine.
Lines 46 -64: Please, revise the paragraphs. There are many repeated information. Summarize the information and show the statements supporting the study's main idea.
Lines 102 - 104. The sentence is confusing. Please, rephrase.
Lines 113 -114: Which are the PCR conditions? Please, show it.
Lines 148 - 149 Were observed differences in autumn and winter? Please rephrase the sentence. The text is tough to read.
Lines 151: This information is repeated.
Figures: They are not available in the manuscript.
I recommend a revision by a native speaker.
Reviewer 2 Report
The manuscript entitled “Environmental factors affect the diversity and composition of environmental microorganisms in Shaoxing rice wine producing area” highlights the role of microbial diversity during the fermentation in different season.
Overall, the manuscript lacks clarity at many points. The significance of the study is not mentioned clearly. In addition, the overall quality of English is poor and requires a thorough grammatical correction and revision. There are many places which make no sense (eg. Line 329-331).
The authors should recheck and ensure that all genus names are italicized.
Finally, after careful examination, I appreciate the novelty and quality of work and recommend adopting suggested improvements.
Specific comments:
Comment 1: line no.s 30-33 and also throughout the manuscript, please write genus name in italics
Comment 2: Please use updated terminology for Lactobacillus throughout the manuscript.
Comment 3: line no. 104 “pretreatment method” please explain briefly.
Comment 4: line no. 109 please correct “DAN” to DNA
Comment 5: please add statistical analysis section and provide the information about all statistical tools used for data analysis
Comment 6: line no. 164 “positively correlated with humidity and O3_8H” O3_8h what it represents? Please mention clearly.
Comment 7: line no. 167-170 “The bacterial communities were significantly correlated with temperature in the four seasons, and were negatively correlated with temperature in spring, autumn and winter, while the opposite was true in summer.” Not clear, please rewrite the sentence.
Comment 9: line no. 173 please recheck and correct NO
Comment 10: line no. 193- 196 sentence too long and unclear. Please rewrite more clearly
Comment 11: line no. 258 please write in Italics “Lactobacillus brevis”
Comment 12: line no. 341 “In these paper, high-throughput sequencing technology was used to analyze the structure and diversity of environmental microbial communities in Shaoxing rice wine producing areas. Please correct or rewrite.
Comment 13: The conclusion is too long. Please reduce, highlighting the major findings and future perspectives of the study.
Extensive review and improvement in English required.
